# Morphological Characteristics and Expression Patterns of *CmCYC2c* of Different Flower Shapes in *Chrysanthemum morifolium*

**DOI:** 10.3390/plants12213728

**Published:** 2023-10-30

**Authors:** Taijia Qiu, Song Li, Kunkun Zhao, Diwen Jia, Fadi Chen, Lian Ding

**Affiliations:** 1National Key Laboratory of Crop Genetics & Germplasm Enhancement and Utilization, Nanjing Agricultural University, Nanjing 210095, China; 2022804223@stu.njau.edu.cn (T.Q.); 2018104092@njau.edu.cn (S.L.); 2016204028@njau.edu.cn (K.Z.); jiadiwen@stu.njau.edu.cn (D.J.); chenfd@njau.edu.cn (F.C.); 2Key Laboratory of Flower Biology and Germplasm Innovation, Ministry of Agriculture and Rural Affairs, Nanjing Agricultural University, Nanjing 210095, China; 3Key Laboratory of Biology of Ornamental Plants in East China, National Forestry and Grassland Administration, Nanjing Agricultural University, Nanjing 210095, China; 4Zhongshan Biological Breeding Laboratory, No. 50 Zhongling Street, Nanjing 210014, China; 5College of Horticulture, Nanjing Agricultural University, Nanjing 210095, China; 6Key Laboratory of Landscaping, Ministry of Agriculture and Rural Affairs, Nanjing Agricultural University, Nanjing 210095, China

**Keywords:** inflorescence forms, flat petal, spoon petal, tubular petal, dorsal petal, floret development, *CYCLOIDEA* gene

## Abstract

The chrysanthemum is widely used as a cut flower, potted flower, and garden flower worldwide and has high ornamental, edible, and medicinal value. The flower heads, composed of ray florets and disc florets, are the most diverse in terms of morphology among ornamental plants. Here, we compared and analyzed the developmental processes of different capitulum types as well as ray florets and disc florets. Morphological differentiation of the two florets occurred on the dorsal domain of the petals at stage Ⅳ of flower development, and differences in stamen development occurred at stage Ⅴ. The dorsal domain of the ray florets and the early stage of flower development were also an essential site and period, respectively, for the differences among capitulum types. In situ hybridization revealed that *CmCYC2c*, whose homologs are involved in the specification of floret identity in Asteraceae, was expressed in both the dorsal and ventral domains of the ray petals in the tubular-type chrysanthemum, whereas, it was differentially transcribed in the ray petals of flat- and spoon-type chrysanthemum cultivars and had lower or no expression in the dorsal domain and higher expression in the ventral domain at stage Ⅳ. Our study indicates that the expression pattern of *CmCYC2c* on the dorsal domain of the ray floret at stage Ⅳ contributes to the formation of diverse flower head types in chrysanthemums.

## 1. Introduction

The chrysanthemum is widely used as a cut flower, potted flower, and garden flower worldwide and has high ornamental, edible, and medicinal value [1,2]. Like other Asteraceae plants, chrysanthemums have a typical capitulum consisting of peripheral female ray florets and central bisexual disc florets [3]. The different florets on the capitulum give the colorful chrysanthemum a rich variety of inflorescence forms [4], which is deeply loved by people. The inflorescence form is an important ornamental feature of the chrysanthemum, and the shape and number of florets are the main factors contributing to the formation of different inflorescence forms in the chrysanthemum [5]. Generally, the disc florets of chrysanthemums can be divided into two types: the anemone type and the non-anemone type [6]. The differences between the two types are only manifested in the coloring and petal elongation [6]. The ray florets of the chrysanthemum have extremely rich forms and are also the main factor involved in forming different capitula types [7]. The corolla splitting of the ray floret, which is later defined as the corolla tube merged degree (CTMD) [8], is considered to be the basis for the classification of ray florets [9]. According to the CTMD, ray florets are divided into three main types: flat, spoon, and tubular [8]. Other complex petals are formed based on these three basic shapes, such as those with irregular edges and/or appendages of the ray floret corolla which, in turn, leads to various flower forms in chrysanthemums [2,10]. Due to the diverse petal types and rich colors of chrysanthemums, many countries have established classification standards for chrysanthemums [11]. The National Chrysanthemum Association of the United States classifies chrysanthemums into 13 basic types [12]. Despite some overlap, Chinese Horticultural Society has conducted a more precise classification of chrysanthemum flower forms. According to the classification criteria of the Chinese Horticultural Society developed in 1983 [13], chrysanthemums are divided into five petal types: flat, spoon, tubular, anemone, and irregular. Based on this, 30 inflorescence forms are expanded. The inflorescence and floret morphogenesis were detailed and monitored in *Chrysanthemum lavandulifolium,* which is an ancestral diploid wild species of *Chrysanthemum morifolium* with only one whorl flat ray floret [14]. Morphological differences between the flat and tubular petals have been analyzed in *Chrysanthemum vestitum*, which is a major origin species of cultivated chrysanthemum [7]. Our previous study revealed hooked petal morphogenesis compared with straight tubular petals using SEM and a histological analysis in *Chrysanthemum morifolium* [15]. Song et al. recorded detailed observations and measurements of ray florets of different petal types in large-flowered chrysanthemums at the full-bloom stage and classified them through quantitative indicators [8]. In summary, the reported studies have mainly focused on the single petal form, the late stage of flower development, and the diploid wild species of *Chrysanthemum morifolium*, while a comprehensive morphological analysis of various ray florets at the early stage of flower development in *Chrysanthemum morifolium* is lacking.

Lots of differentially expressed genes (DEGs) involved in the petal form regulation of ray florets were screened by a transcriptome analysis of flower buds and flower organs in *Chrysanthemum morifolium* [16]; flat, spoon, and tubular ray florets in *Chrysanthemum vestitum* [7]; and six key stages of capitulum development in *C. lavandulifolium* [7,14]. These genes include MADS-box genes, planthormone-related genes, AP2/ERF family genes, and *CYCLOIDEA*-like genes. Moreover, the expression patterns of essential candidate genes, such as *ClPIN1*, *ClARF5*, *CmYAB1*, and *CmCYC2s* etc., were verified by RT-PCR and RNA in situ hybridization [14,15,16]. In recent years, the CYC-like TCP domain proteins have become a research hotspot for floret development in Asteraceae [10]. The CYC group experienced repetition and generated three main groups: CYC1, CYC2, and CYC3 [17]. *CYC2-like* genes are considered to be important factors in identifying the dorsal petals and are differentially expressed in the dorsal and ventral petals [18,19]. In addition, a series of studies showed that the *CYC2-like* genes were specifically expressed in the ray florets of the Asteraceae and determined the identity of the ray florets. In *Helianthus annuus* (sunflower), the *CYC2* homologous gene *HaCYC2c* is a powerful candidate gene for regulating the identity of ray florets [20], and the overexpression of *HaCYC2c* results in a capitulum with only ray florets [21]. In *Gerbera hybrida*, *GhCYC2* is expressed in the marginal ray florets, but not in the central disc florets [22]. Moreover, the overexpression of *GhCYC2* in *Gerbera hybrida* can lead to the disc florets gaining the identity of the ray florets [22]. Other *CYC2-like* genes have redundant functions in the regulation of the ray floret identity but have diversified functions in the regulation of stamen development and petal growth in *G*. *hybrida* [23]. The *CYC2* homologues *RAY1*, *RAY2*, and *RAY3* in *Senecio vulgaris* were found to be specifically expressed in ray florets [24,25]. The *Gaillardia* species has two different ray petal shapes, ligulate and funnel-shaped. After *CYC2c* gene silencing in *Gaillardia*, the ray florets converted from ligulate to funnel-shaped [26]. In the study of *Chrysanthemum lavandulifolium*, the overexpression of *CmCYC2c* resulted in significant increases in the number and petal length of ray florets [27]. *CYC2s* also have an effect on the morphogenesis of the entire capitulum. ClCYC2g may be an important cause for the shift from a radiate to a disciform capitulum [28]. At the floret level, *Cyc2CL-1* and *Cyc2CL-2* regulate both petal and stamen development in ray florets in the chrysanthemum [29]. Further research on *CYC* genes in different petal forms will strengthen our understanding of flower shape development in Asteraceae.

In this study, the scanning electron microscopy, light microscopy, and paraffin section methods were used to compare the development processes of different types of chrysanthemum flower heads, as well as ray florets and disc florets, and to explore the significant parts and developmental stages of the morphological differences in the capitula. In addition, the spatiotemporal expression pattern of *CmCYC2c* in different inflorescence types was investigated by RNA in situ hybridization to explore its role in inflorescence morphogenesis. 

## 2. Results

### 2.1. Structural Aspects of Ray Floret and Disc Floret Development 

The capitulum of the chrysanthemum is composed of zygomorphous female ray florets and actinomorphic bisexual disc florets, which are the most typical characteristics of plants in the Asteraceae. To explore the differences between ray and disc florets in the early development stages, morphological analysis was carried out using paraffin sections and scanning electron microscopy (Figure 1 and Appendix A). Taking the double flowers and spoon petal variety ‘Jinba’ as an example, the organogenesis of two types of florets was monitored. After transition to flowering, the shoot apex shifted from a shoot apical meristem (SAM) to an inflorescence shoot apical meristem (IM). The involucre primordia were produced on the side of the hemispherical IM (Appendix A). Dome-shaped flower primordia were initiated from the axil of involucre primordia, and this was the first stage of flower development. After several whorls of ray floret primordia initiation, the disc floret primordia appeared in the center part of the inflorescence, and there was no difference in morphology compared with the ray flower primordia at stage Ⅰ (Figure 1A,F). At stage Ⅱ, the petal primordia were beginning to be visible in both dorsal and ventral domains without morphological differences resulting in ring-shaped petal primordia (Figure 1B,G). At stage Ⅲ, petal primordia development continued, and the stamen primordia started to form in both the ray florets and the disc florets (Figure 1C,H). This occurred until stage Ⅴ, when the stamen primordia stopped developing in the ray florets, whereas further growth could be achieved in the disc florets. At stage Ⅳ, the ray petal became distinct from the disc petal. The ventral domain of the petal primordia elongated at a higher rate than the dorsal domain of the petal primordia, resulting in longer petal primordia in the ventral domain and covering half of the flower bud in the ray floret at stage Ⅳ (Figure 1D), whereas in the disc florets, the dorsal and ventral domains of the petal primordia developed synchronously and grew to completely cover the floral bud (Figure 1I). The carpel primordia arose at this stage. Flower organs continued to grow at stages Ⅴ and Ⅵ, except for the stamen primordia in the ray florets. At this time, the ray florets became female ones with bilateral symmetrical petals, and the disc florets developed into bisexual flowers with radially symmetrical petals.

### 2.2. Morphological Observations of Different Inflorescence Forms of the Chrysanthemum

To clarify the morphological differences among various inflorescence and petal forms, a total of 12 chrysanthemum cultivars in five categories were observed at the full-bloom stage. As mentioned above, the classification of petal types is often based on the CTMD [9,10]. Generally speaking, the larger the merged degree, the more tubular the petals and the smaller the merged degree, the flatter the petals [9]. Group I was the flat-petal type, which has the lowest fusion degree in the corolla tube, where only the base merges into a tube, and the merged portion accounts for less than one-fifth of the total petal length (Figure 2) [8]. The morphology of the petals is also different among the flat petal varieties (Figure 2). For example, the ray petals of ‘Jinbeidahong’ are incurvate, and the ray petals of ‘Fengtingzhuangyuan’ are twisty and evaginable (Figure 2A,B). The spiral and straight ray petals are displayed in the ‘Chifengxuanwei’ and ‘Baipingpang’ cultivars, respectively (Figure 2C,D). Group II was the spoon ones, where the base of the petals merges into a tube, and the upper part unfolds into a spoon shape with the fusion area being between one-fifth and three-fifths of the total petal length (Figure 3) [8]. In Group III, the corolla tube is highly fused, forming a tubular petal, and more than three-fifths of the portion is merged (Figure 4). Slender tubular petals and thick, straight tubular petals appeared in ‘Jinsongyue’ and ‘Guohuahuabaihe’ (Figure 4A,B). Some tubular ones formed hook-like structures on the tips of ray florets (Figure 4C). Group IV was the one with irregular petals. This group has multiple degrees of fusion, but the distal corolla splits into various shapes (Figure 5). The irregular petals are divided into three types: the unguiculate type, which are shaped like the claws of a dragon (unguiculate) (Figure 5A); the chenille-like shape, where the corolla splits into several portions at the distal domain (Figure 5B); and aristate ones, which have some burr-like structures on the abaxial surface of the corolla (Figure 5C). Unlike the first four groups, group Ⅴ was classified based on the morphology of the disc florets and referred to those with elongated disc florets, also called the anemone type. In contrast to the non-anemone types, the petals of the disc florets in the anemone group are elongated, forming a polygonal corolla (Figure 6). The petal length exceeds, or is slightly shorter than, the pistil length in anemone types, so when in full bloom, the pistils and stamens are hidden within the petals (Figure 6). The disc florets of non-anemone types are yellow and the petals do not extend significantly, resulting in a protruding stigma and visible mature stamens in the capitulum. These observations indicate that different inflorescence types have great morphological differences. In addition to the CTMD, the morphology of the distal of petals is also a key factor in determining the flower forms.

### 2.3. Comparison of the Early Development of Different Capitulum Types

In order to further determine the critical stages and parts of different capitulum types, the morphologies of four groups of flower heads with different petal types were observed during the early developmental stage using SEM and light microscopy. One representative cultivar was selected for each of the flat, spoon, and irregular types, and two representative cultivars were selected for the tubular type (Figure 7). The results showed that the ring-shaped petal primordia initiated in all inflorescence types at stage Ⅱ (Figure 7B,N,S). And, at stage Ⅳ, the dorsal domain of the petal primordia lagged behind the ventral domain to varying degrees in the flat type, the spoon type, and the hooked petal tubular type (Figure 7C,G,P). In the unguiculate irregular type and straight tubular type, the dorsal and ventral growth was synchronized, despite the minimal splits at the distal end in the unguiculate irregular type (Figure 7I–K,R–U). The differences were more pronounced in the dorsal domain as the florets grew (Figure 7D,H,L,Q,U). Although the tubular type had the highest degree of fusion, the flat type had the lowest degree of fusion. However, the dorsal domain of the hooked petal (one of the tubular flower types) was not fully fused (Figure 7M–Q). Our results suggest that the early stage (stage Ⅳ) of flower development is a critical period for petal type formation in the chrysanthemum. The growth of the dorsal domain of the petal determined the flower type. And, the morphology of the distal region of the ray petal was also determined at the early stage of flower development.

### 2.4. Expression Pattern Analysis of CmCYC2c in Different Inflorescence Types of Chrysanthemum

It was proved that *CmCYC2c* functions as a predominant regulator in ray petal development and is differentially expressed between ray florets and disc florets [16,27]. To further clarify the spatiotemporal expression patterns of *CmCYC2c* in different types of ray flowers, RNA in situ hybridization was performed in the flat-type and tubular-type chrysanthemum cultivars. *CmCYC2c* was transcribed in the ray petal primordia when it was initiated at stage Ⅱ in both the flat-type and tubular-type chrysanthemum cultivars (Figure 8A,E). At stage Ⅲ, *CmCYC2c* was also expressed in the initiating stamen primordia, and there was little difference in the expression pattern of *CmCYC2c* between the dorsal and ventral domains of the ray flower in the tubular type ‘Yellow Beehive’ (Figure 8B). At stage Ⅳ, the signals of *CmCYC2c* were located in the distal region of the dorsal and ventral domains of the ray flower in tubular-type chrysanthemum‘Yellow Beehive’ (Figure 8C). Whereas the expression of *CmCYC2c* was attenuated in the dorsal domain of the ray petal in the spoon type ‘Jinba’ at stage Ⅲ, although it continued to be expressed in the ventral domain (Figure 8F). And, the expression of *CmCYC2c* was almost undetectable in the dorsal domain of the ray petal primordia at stage Ⅳ in ‘Jinba’ (Figure 8G). The expression of *CmCYC2c* was weaken in the attenuated stamen primordia and was displayed in the carpel primordia at stage Ⅳ (Figure 8C,G). Moreover, the dorsal–ventral expression pattern was also displayed in another tubular-type cultivar ‘White Anna’ (Figure 8D), and the ventral expression pattern was found in the flat cultivar ‘Baipingpang’ (Figure 8H). These results indicate that the expression pattern of *CmCYC2c* differs among cultivars with different petal shapes, especially in the dorsal domain of ray petals, consistent with our previous observation that the dorsal domain is an essential site for morphological differences across various flower types. In summary, the differential expression patterns of *CmCYC2c* in various petal shapes prove that *CmCYC2c* is involved in the morphogenesis of ray floret petals, resulting in the formation of rich petal shapes and different capitulum forms based on this.

## 3. Discussion

### 3.1. Whether the Dorsal Domain of the Ray Florets Elongates during Early Flower Development Is the Main Factor for Flower Shape Diversity in the Chrysanthemum

The capitula of the Asteraceae are typically characterized by individual florets arranged in a spiral pattern on the receptacle, the number of which follows the well-known Fibonacci sequence [28,30,31]. The entire structure is surrounded by sepal-like inner convoluted bracts with protective functions. However, the initiation of the phyllary, ray florets, and disc florets does not occur acropetally. Studies on *C. lavandulifolium* and *Chrysanthemum vestitum*, which have a single whorl of ray florets, showed that disc floret primordia appeared anteriorly to the ray floret primordia [7,14]. The same initiation order was found in a single whorl of the ray floret chrysanthemum cultivar (Figure 7E–H) in this study. But, the second whorl of the ray floret primordia started prior to the inner disc floret primordia and the first whorl of the ray floret primordia in double ray floret chrysanthemum cultivars (Figure 7 and Figure 9A). Moreover, the disc floret primordia appeared after several whorls of ray petals, not after all ray petals (Figure 7 and Figure 9A). These results indicate that the initiation pattern of the floret primordia differs among various capitula in Asteraceae. It appeared that the florets in the second whorl are initiated first, regardless of whether the second whorl is disc or ray florets.

The ray petal primordia began and were ring-shaped at stage Ⅱ in different petal-type chrysanthemum cultivars (Figure 7 and Figure 9). After that, the unsynchronized dorsal and ventral growth resulted in the dorsal domain of the petals being shorter than the ventral domain of the petals, resulting in bilateral symmetrical flat and spoon petals (Figure 7 and Figure 9). In contrast, dorsal and ventral synchronous development resulted in radially symmetric tubular types (Figure 7 and Figure 9). Previous research on different ray petal types of *Chrysanthemum vestitum* suggested that the stagnation of the dorsal domain of the petals leads to the formation of flat petals, and this difference mainly occurs during the initial stage with vigorous cell division [7]. Thus, the development of the dorsal domain of the petals of the ray floret at the early stage determines the ray petal type in chrysanthemums. A spatial transcriptomics analysis of ray florets at this stage will further screen the crucial genes and networks in the regulation of petal shape development.

Our morphological analysis of ray florets and disc florets suggested that the morphological differences between the two types of florets also occurred on the dorsal domain of the petals, similar to the differences between flat and tubular ray petals. It has been suggested that the bilaterally symmetrical ray florets of Asteraceae evolved from radiologically symmetrical disc florets, and after multiple evolvements, ray florets with abundant morphologies formed [32]. In the non-anemone chrysanthemum cultivars, the disc florets are yellow, and the petals do not elongate; in the anemone type, the disc florets are colorful and have elongated petals, being similar in morphology to the tubular ray florets. Therefore, we speculate that the disc florets, especially the disc florets of the anemone type, and the ray florets may have similar regulatory mechanisms.

The different types of ray florets and disc florets, especially the diversiform ray florets, form rich capitulum types in chrysanthemums. The corolla tube merged degree (CTMD) is considered to be the basis for differences in ray petal shapes [33], while the type of disc floret can be roughly summarized as the anemone type or non-anemone type. Song et al. provided a more accurate definition of the CTMD in their extensive observations of traditional Chinese chrysanthemums, confirmed grading standards, and provided a detailed classification by quantifying the characteristics of chrysanthemum petals [8]. Our results illustrate that, besides the CTMD, the degree of fusion at the distal end of ray petals is also an important factor affecting the petal morphology.

### 3.2. CmCYC2c Is Responsible for the Morphogenesis of Flat, Spoon, and Tubular Ray Petals 

*CYCLOIDEA* (*CYC*), a TCP family member, was first identified as a key transcription factor that regulates floral symmetry [34]. In Antirrhinum, *CYC* is only expressed in the elongated dorsal petals but not in the non-elongated ventral domains of petals, indicating that the spatial distribution of *CYC* is closely related to the flower symmetry [34]. In addition, a comparison of the flower forms between *Arabidopsis thaliana* and *Iberis amara* revealed that the dorsal petals of *Iberis amara* continuously elongate to form a bilateral symmetry due to the continuous expression of *IaTCP* (*CYC*), while *AtTCP1* (*CYC*) is only expressed in the early stage of the Arabidopsis flower primordia, resulting in uniform growth on the dorsal and ventral sides [35,36]. In Compositae, *CYC-like* genes have undergone several gene duplication events [17]. Among them, CYC2 clade genes played an important role in regulating the floret identity [20,21,22,37]. There are six members of the CYC2 clade in the chrysanthemum, with the highest expression of *CmCYC2c* occurring in the corolla of ray florets [27]. The overexpression of *CmCYC2c* in *Chrysanthemum lavandulifolium* leads to increases in the flower number and ray petal ligule length. Knocking down the expression of *ClCYC2g* (homologous with *CmCYC2c*) in *Chrysanthemum lavandulifolium* converts the ray florets to the disc-like form. These findings indicate that the expression pattern of *CYCs* is critical for their functions. In this study, we found that *CmCYC2c* was transcribed in ring-shaped petal primordia at stage Ⅱ, when the petal primordia initiated (Figure 8A,E and Figure 9). However, *CmCYC2c* was differentially expressed in the dorsal and ventral domains of the ray petals of the flat- and spoon-type chrysanthemum cultivars, having lower or no expression in the dorsal domain and higher expression in the ventral domain at stage Ⅳ (Figure 8G,H and Figure 9B). There was no difference in the expression of *CmCYC2c* between the ventral and dorsal domains of the ray petals in the tubular type, and it was highly expressed in both the dorsal and ventral domains of petals at stage Ⅳ (Figure 8C,D and Figure 9). Our results indicate that the sustained expression of *CmCYC2c* in the early stage could promote petal elongation and indirectly affect the petal symmetry. Morphological differences between the flat and ray floret types occurred during the early stage, similar to in *C. vestitum*. This is an active period of cell division [7]. Thus, *CmCYC2c* may regulate the petal length by promoting cell division in Chrysanthemum.

*CmCYC2c* is highly expressed in the ray florets of both flat- and tubular-type chrysanthemums and has low expression in the disc florets [16,27]. Given that the petals of the ray florets are significantly elongated, while the petals of the disc florets are relatively short, high *CmCYC2c* expression in the ray florets promotes the elongation of the ray petals, while inadequate expression of *CmCYC2c* limits petal elongation in the disc florets. However, the molecular mechanisms responsible for the differential expression of *CmCYC2c* between the ray florets and disc florets is still unknown. *CYC-like* genes could form homodimers or heterodimers, and these dimers could bind to the promoter of the *CYC-like* genes themselves and activate their own expression, thereby forming a positive feedback loop and enabling the continuous expression of *CYC-like* genes [38,39]. Similar regulation mechanisms were partially revealed in chrysanthemums. Y2H and BiFC assays proved that CmCYC2b interacts with CmCYC2c, CmCYC2d, and CmCYC2e [16,40]. CmCYC2c can promote the transcription of *ClCYC2f* by binding to its promoter, and the expression of *ClCYC2f* is upregulated in the *CmCYC2c* overexpression line in *C. lavandulifolium* [27,40]. The SEP-like protein GRCD5 targets the *GhCYC3* gene and activates *GhCYC3* expression during ligule floret development in *Gerbera hybrida* [41]. However, the expression domain of *GRCD5* extends to the disc flowers and is larger than that of *GhCYC3*. Therefore, the direct factors that specifically inhibit the expression of *CYC2-like* genes are still unclear. Having a more precise sampling period and sampling domain will be helpful to solve this problem. 

The asymmetrical expression of *CmCYC2c* led to the development of ray florets with flat- and spoon-shaped petals, while continuous symmetrical expression resulted in tubular ray florets (Figure 9). A low level of *CmCYC2c* expression produced disc florets (Figure 9). And, the distinct morphologies of ray and disc florets contributed to the formation of various capitulum types. The different spatial distributions of *CmCYC2c* within the inflorescence gave rise to a diverse array of flower shapes. At stage Ⅳ of flower development, the expression of *CmCYC2c* varies in different flower shapes. By changing the expression pattern of *CmCYC2c* by gene editing and other molecular methods, the target flower shapes will be obtained. Our findings regarding the spatiotemporal expression patterns of *CmCYC2c* in different types of ray flowers, particularly the variations in the dorsal and ventral domains of ray petals provides a basis for further understanding the genetic control of petal morphogenesis and its role in shaping the diversity of flower types in chrysanthemums.

## 4. Materials and Methods

### 4.1. Plant Materials

Cuttings of chrysanthemum cultivars with different inflorescence forms were obtained from the Nanjing Agricultural University Chrysanthemum Germplasm Resource Preservation Center (Nanjing, China) and grown in a greenhouse using standard management practices. The cultivars used in this study were colonized in late July and entered full bloom in early November with the temperature in the greenhouse ranging from 30 °C to 18 °C. Flower buds of different sizes, florets, and inflorescences at full bloom were used in subsequent experiments.

### 4.2. Morphological Observations and Electron Microscopy Scanning

Different capitulum sizes (<2 mm, 2 mm, 4 mm, 6 mm, 8 mm in diameter) were obtained from five chrysanthemum cultivars, and the involucres were removed. Morphological observations were performed using light microscopy (Leica S8 APO, Wetzlar, Germany). The removed involucre inflorescences were fixed in 2.5% *v*/*v* glutaraldehyde. After critical point drying and coating with gold, scanning electron microscopy was conducted using an SU8010 device (Hitachi, Tokyo, Japan). At the full-bloom stage, the Canon 90D camera was used to record the morphology of inflorescence in two views: side and top. The ray florets from the outer to the inner whorls and anemone disc florets were photographed to record the petal shapes.

### 4.3. Paraffin Sections

Whole inflorescences of the chrysanthemum ‘Jinba’ were sampled when the inflorescences were <2mm, 2 mm, 4 mm, and 8 mm in diameter. After fixation, embedding, sectioning, and dewaxing, as described in our previous study [16], staining was performed with 0.1% (*m*/*v*) fast green FCF. Sections were photographed using a DM1000 microscope (Leica Camera AG, Wetzlar, Germany).

### 4.4. RNA In Situ Hybridization

The tubular chrysanthemum cultivars ‘Yellow Beehive’ and ‘White Anna’ and the flat cultivars ‘Jinba’ and ‘Baipingpang’ were used for in situ hybridization. When the diameters of the capitula were 2–3 mm and 4–6 mm, they were sampled and fixed in 3.7% FAA (formalin acetic acid alcohol), dehydrated through an ethanol series, and embedded in paraffin. *CmCYC2c*-specific probes were amplified using primers including the SP6 and T7 RNA polymerase binding sites. *CmCYC2c* probes were labeled through a PCR reaction using the DIG RNA labeling kit (SP6/T7) (Roche, Switzerland). RNA in situ hybridization was performed as described elsewhere [42]. The primer sequences used for probe synthesis are listed in Appendix A.

## 5. Conclusions

In this study, we found that at stage Ⅳ of flower development, the ray and disc florets showed morphological differences, mainly on the dorsal domain of the ray petal. At stage Ⅴ, the stamen primordium of the ray floret was degraded, while the stamen primordium of the disc floret developed normally. Similarly, the dorsal domain of the ray floret at stage Ⅳ of flower development showed the main differences in site and period among the chrysanthemums with different flower types. The expression pattern of *CmCYC2c* differed between flat-type and tubular-type chrysanthemums. In the tubular types, *CmCYC2c* sustained a symmetrical expression in the dorsal and ventral domains of the ray petals, whereas in the flat types, there was almost no expression in the dorsal domain of the ray petals and there was expression in the ventral domain at stage Ⅳ. To sum up, *CmCYC2c* is involved in the formation of various flower forms in chrysanthemums. Our study not only provides new insights into the development of different capitulum types, but also complements the function of *CmCYC2c* in Asteraceae.

## Figures and Tables

**Figure 1 plants-12-03728-f001:**
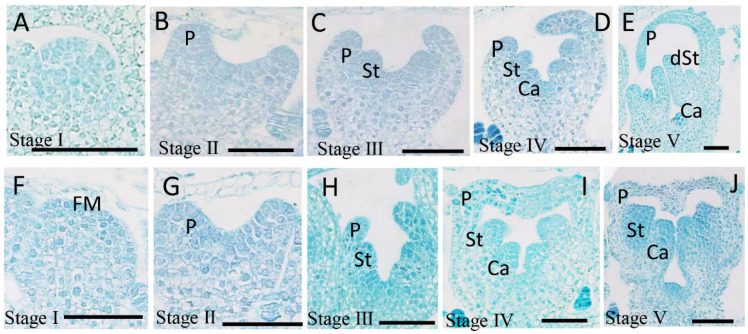
The development process of ray florets and disc florets in ‘Jinba’: (**A**–**E**) ray floret. (**F**–**J**) disc floret. (**A**,**F**) The first stage of floret development when the flower meristem began to appear on the inflorescence meristem (IM). (**B**,**G**) The second stage, the emergence of the petal primordia. (**C**,**H**) The third stage, the initiation of the stamen primordia. (**D**,**I**) The fourth stage, the carpel primordia began to be visible. The petal primordia of the ray florets grew asymmetrically (**D**), whereas the petal primordia of the disc florets grew symmetrically (**I**). (**E**,**J**) The fifth stage, when the stamen primordium of the ray floret was degraded (**E**), while the stamen primordium of the disc floret was normally developed (**J**). P, petal primordium; St, stamen primordium; Ca, carpel primordium; Bar = 100 µm.

**Figure 2 plants-12-03728-f002:**
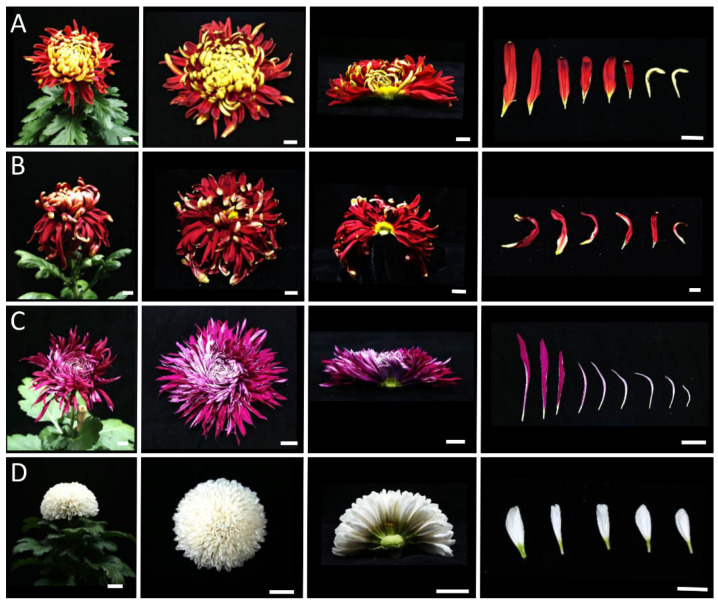
Morphological observations of the flat-type chrysanthemum cultivars at the full-bloom stage. The petals are only fused at the base. (**A**) Whole and half images of the inflorescences and ray florets of ‘Jinbeidahong’. The ray petals are expanded flat in the outer whorls and curved in the inner whorls. (**B**) The inflorescences and ray florets of ‘Tingfengzhuangyuan’. The ray petals are curved (inner whorls) and evaginable (out whorls). (**C**) ‘Chifengxuanwei’. The outer florets are flat, while the inner florets show different degrees and angles of bending. (**D**), ‘Baipingpang’. The petals are folded inwards. The right panel shows ray florets from the outside to the inside whorls. Bar = 2 cm.

**Figure 3 plants-12-03728-f003:**
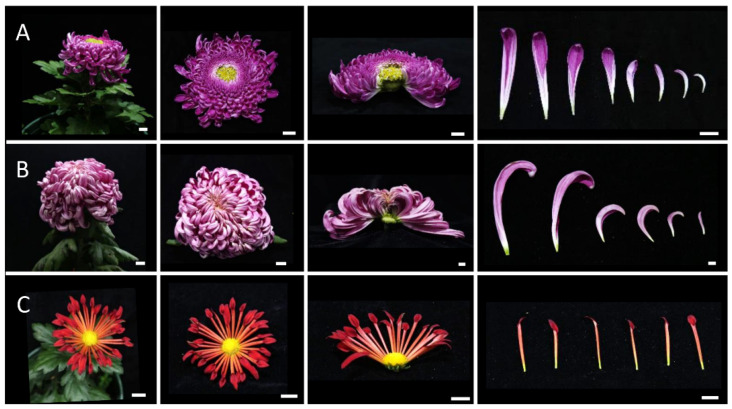
Different spoon-like chrysanthemum cultivars. Whole and half images of the inflorescences and ray florets of ‘Hengshuiyingyue’ (**A**), ‘Guohuaqiangda’ (**B**), and ‘Q5-5’ (**C**). The fusion area of the petal was between one-fifth and three-fifths of the total petal length. Most of the ray flowers of the spoon-type chrysanthemum were spoon petals, and there were also a few flat petals and tubular petals. Bar = 2 cm.

**Figure 4 plants-12-03728-f004:**
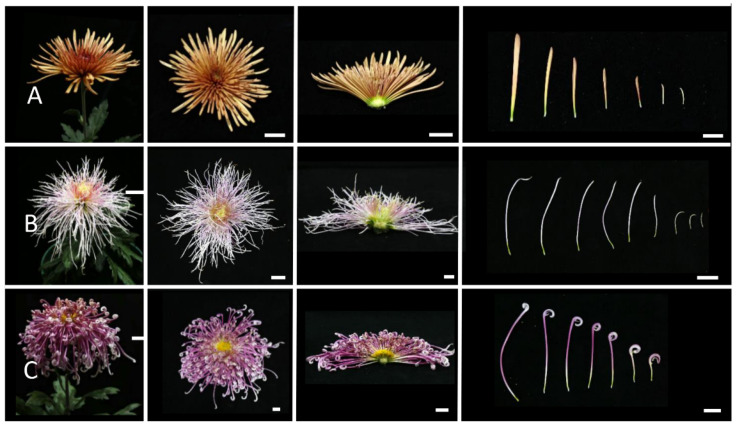
The tubular-type chrysanthemum cultivars. The petals were highly fused. Whole and half images of inflorescences and ray florets of ‘Jinsongyue’ (**A**), ‘Guohuahuabaihe’ (**B**), and ‘Jierilihua’ (**C**). The tubular petals were thick in (**A**), thin and long in (**B**), and hooked at the tip in (**C**). Bar = 2 cm.

**Figure 5 plants-12-03728-f005:**
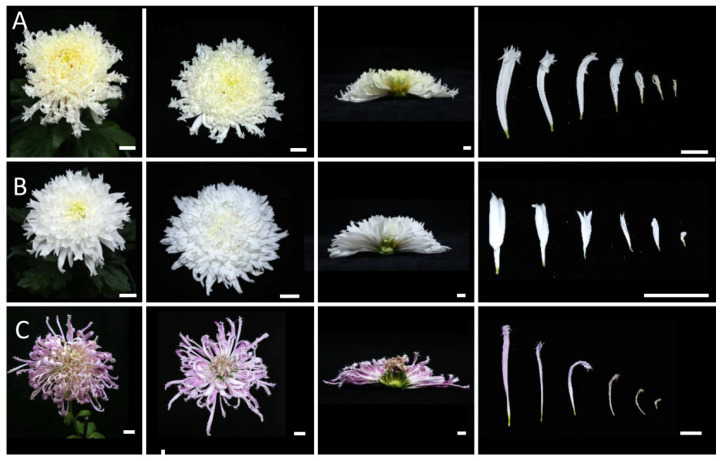
Three different types of irregular-type chrysanthemums. Whole and half images of inflorescences and ray florets of ‘Yinlongfenshui’ (**A**), ‘Qinhuaichunxue’ (**B**), and ‘Danfenmaoci’ (**C**). (**A**) The unguiculate type, in which the corolla is shaped like the claw of dragon. (**B**) The chenille-like type, where the petals are splitting at the tips. (**C**) The aristate one, where a few burr-like structures are found on the corolla. Bar = 2 cm.

**Figure 6 plants-12-03728-f006:**
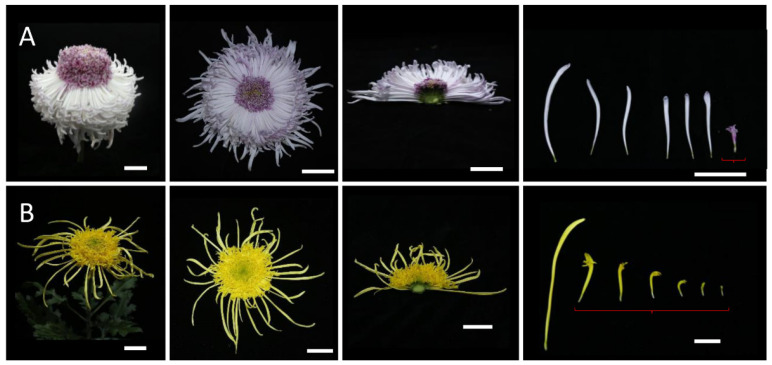
Anemone-type inflorescences and florets. Whole and half images of inflorescences and ray florets of ‘Oufentuogui’ (**A**), and ‘Yuemingxingxi’ (**B**). (**A**) The petals of the disc florets were elongated and splitting at the tips and formed an anemone-like structure in the center of the capitulum. (**B**) The petals of the disc florets varied in length but were all longer than both the pistils and the stamens, so both were hidden in the petals. Bar = 2 cm.

**Figure 7 plants-12-03728-f007:**
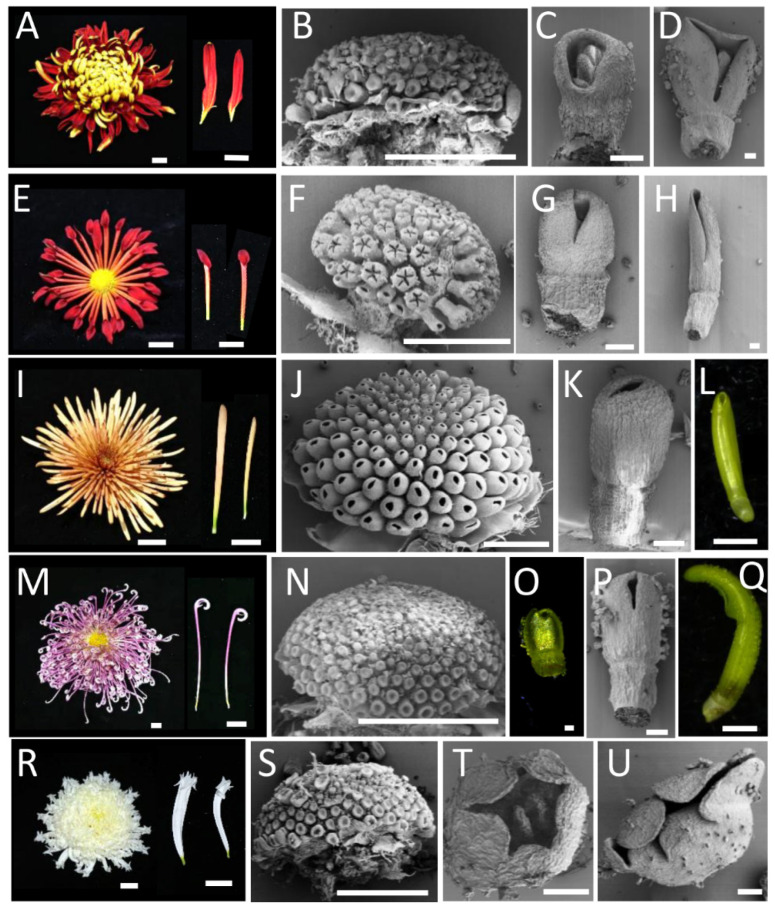
Morphological comparison of different capitulum-type chrysanthemums at the early stage of flower development. The capitula and florets of the flat type ‘Jinbeidahong’ (**A**–**D**), spoon type ‘Q5-5’ (**E**–**H**), tubular type ‘Jinsongyue’ (**I**–**L**) ‘Jierilihua’ (**M**–**Q**), and irregular type ‘Yinlongfenshui’ (**R**–**U**). (**C**,**D**) The scanning electron microscopy results of the flat petal chrysanthemum showed that the length of the dorsal domain of the petal was shorter than that of the ventral domain of the petal, forming an unfused flat petal. (**G**,**H**) The degree of fusion of the spoon type was higher than that of flat type, and there was still a large split in the petal primordium. (**K**,**L**) The petals were highly fused, forming a tube. (**O**–**Q**) The hooked petals were slightly non-fused in the dorsal domains of the petals at the early stage and formed hooks at the late stage. (**T**,**U**) The ray floret primordia of the irregular type showed several splits at the tips of the petals at the early stage. (**A**,**E**,**I**,**M**,**R**) Bar = 2 cm. (**B**,**F**,**J**,**L**,**N**,**Q**,**S**) Bar = 1 mm. (**C**,**D**,**F**–**H**,**J**,**K**, **N**–**P**,**S**–**U**) Bar = 100 μm.

**Figure 8 plants-12-03728-f008:**
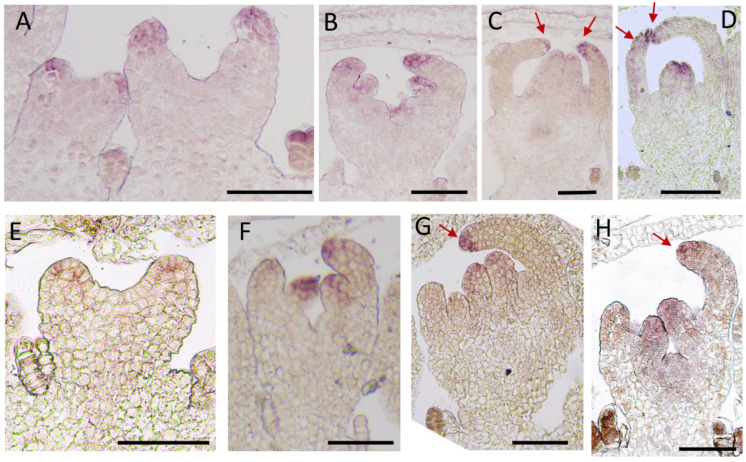
RNA in situ hybridization analysis of *CmCYC2c* in different flower shapes. (**A**–**C**) The tubular type ‘Yellow Beehive’. (**A**) The *CmCYC2c* transcripts were present in the petal primordia at stage Ⅱ. (**B**) At stage Ⅲ, *CmCYC2c* was transcribed in both the distal region of the ventral and dorsal domains of the petal primordia and stamen primordia with a slightly reduced abundance in the dorsal domain of the petal. (**C**) At stage Ⅳ, *CmCYC2c* signals were visible in the stamen and carpel primordia, and the expression of *CmCYC2c* was detected in both the ventral and dorsal domains of the petal primordia. (**D**) Another tubular cultivar, ‘White Anna’, which showed a similar symmetrical expression pattern for *CmCYC2c* to the ‘Yellow Beehive’. (**E**,**F**) The flat cultivar ‘Jinba’. (**E**) *CmCYC2c* transcripts were detected in the petal primordia at stage Ⅱ. (**F**) At stage Ⅲ, *CmCYC2c* transcripts appeared in the petal and stamen primordia, while the abundance was decreased in the dorsal domain of the petal. (**G**) At stage Ⅳ, *CmCYC2c* was transcribed in the carpel primordia, and the transcripts in the stamen primordium were reduced. And, the transcripts in the ventral petals were significantly visible and there were almost none in the dorsal domain of the petals. (**H**) Another flat cultivar ‘Baipingpang’ displayed a similar asymmetrical expression pattern to *CmCYC2c* with ‘Jinba’. The arrows indicate the symmetrical (**C**,**D**) and asymmetrical (**G**,**H**) expression patterns of *CmCYC2c*. Red arrows indicate the expression domain of *CmCYC2c* in tubular (**C**,**D**) and flat petals (**G**–**H**). Bar = 100 µm.

**Figure 9 plants-12-03728-f009:**
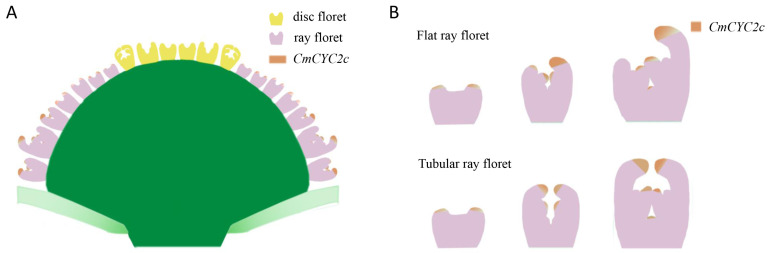
Schematic diagram of the spatiotemporal distribution of *CmCYC2c* in the capitulum (**A**) and flat and tubular ray florets (**B**).

## Data Availability

The datasets supporting the conclusions of this article are included within the article and Appendix A.

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
