# Peer review of "Morphological Characteristics and Expression Patterns of CmCYC2c of Different Flower Shapes in Chrysanthemum morifolium"

_plants, 2023, doi:10.3390/plants12213728_

Round 1

Reviewer 1 Report

Comments and Suggestions for Authors

This study is quite sound experimentally, however, the manuscript requires some improvement.

Title

The title can be improved, as the study is not about “analysis”, but about the results of this analysis.

There is a problem with a few terms in the Abstract, Conclusions, and throughout the text.

Maybe some schematic representation could be helpful, for example in Results or Discussion.

The problem is that the Authors use and mix similar terms:

Ventral dorsal primordia of petals (or florets)

Ventral and dorsal petals

Ventral and dorsal domain of petals

Ventral dorsal region of petals

Abstract

Therefore, in 23-28 lines, two sentences need to be clarified because of mixed terms, starting with “In situ…ventral petal at stage 4".

Results

The legends of the figures need improvement.

Figure 1, 139-147 lines. The description could be shortened by avoiding the repetitive phrase “stage of floret development” (repeated 5 times)

Bar=100um? What is this “um” dimension?

What does the abbreviation "IM" mean?

Figure 2, 183-190 lines. The description could be shortened by avoiding the repetitive phrase “the whole and half inflorescences and ray florets” (repeated 4 times).

Figure 6, 208-21 lines. The authors write “disc florets” (2 times). Perhaps, they mean “ray florets”?

Figure 7, lines 233-242 lines. The images of scanning microscopy are present in C, D (not A-D); G, H (not E-H; K, L (not I-L); O-Q (not M-Q), and T, U (not R-U).

Figure 8, 271-285 lines. The description could be shortened by avoiding the repetitive phrase “the in situ hybridization analysis” (repeated 3 times).

Bar=100nm? Is it true?

Discussion

The discussion is difficult to read in many parts.

In the first part, 3.1, the Discussion greatly overlaps with the Results. Perhaps, some schematic representation would be helpful.

In the second part, 3.2, 363-375 lines, the text is not quite relevant to this study and needs to be modified.

Conclusions

It needs to be thoroughly checked through for the correct use of the terms “dorsal and ventral domains, parts…petals”. An example of mixing these terms is the sentence in lines 418-420, “In tubular types CmCYC2c was sustained symmetric expressed in the dorsal and ventral domain of ray petals, whereas in the flat types, it was almost no expression in the dorsal petals and expression in the ventral petals.”

Comments on the Quality of English Language

Author Response

Comments and Suggestions for Authors

This study is quite sound experimentally, however, the manuscript requires some improvement.

Title

The title can be improved, as the study is not about “analysis”, but about the results of this analysis.

Response:Thanks for your comments and corrected as suggested. We revised the title as “Morphological characteristics and expression patterns of CmCYC2c of different flower shapes in Chrysanthemum morifolium

There is a problem with a few terms in the Abstract, Conclusions, and throughout the text.

Response:We are grateful to this comment and modified the confused terms throughout the text.

Maybe some schematic representation could be helpful, for example in Results or Discussion.

Response:Thanks for your comments and added a schematic diagram in the Discussion.

The problem is that the Authors use and mix similar terms:

Ventral dorsal primordia of petals (or florets)

Ventral and dorsal petals

Ventral and dorsal domain of petals

Ventral dorsal region of petals

Abstract

Therefore, in 23-28 lines, two sentences need to be clarified because of mixed terms, starting with “In situ…ventral petal at stage 4".

Response:We are grateful to this comments and modified the sentences as“In situ hybridization revealed that CmCYC2c, which homologs are involved in specifying floret identity in Asteraceae, was expressed in the both dorsal and ventral domains of the ray petals in tubular type chrysanthemum. Whereas, it was differentially transcribed in the ray petals of flat and spoon type chrysanthemum cultivars, lower or no expression in dorsal domain and higher expression in ventral domain at stage â…£.”in line 24-29

Results

The legends of the figures need improvement.

Figure 1, 139-147 lines. The description could be shortened by avoiding the repetitive phrase “stage of floret development” (repeated 5 times)

Response:Thanks for your comments. We’ve deleted the duplicate “of floret development" in lines 142-146 to keep it concise.

Bar=100um? What is this “um” dimension?

Response:Thank you for your comments, and we're sorry to mismark "μm" as "um" in writing, which has been revised on line 148, thank you very much for the reminder.

What does the abbreviation "IM" mean?

Response:Thank you for your comments. We have added the full name and abbreviation “inflorescence meristem (IM)” on line 142 to facilitate readers to quickly understand the content in the figure.

Figure 2, 183-190 lines. The description could be shortened by avoiding the repetitive phrase “the whole and half inflorescences and ray florets” (repeated 4 times).

Response:Thank you very much for your suggestion, we have shortened or deleted the repetitive phrase "the whole and half inflorescences and ray florets" in lines 186-191 to make it short and concise.

Figure 6, 208-21 lines. The authors write “disc florets” (2 times). Perhaps, they mean “ray florets”?

Response:Thank you very much for your comments. In Figure 6, we focus on the morphology of disc florets in anemone-type chrysanthemum, and it is correct to use "disc florets" here.

Figure 7, lines 233-242 lines. The images of scanning microscopy are present in C, D (not A-D); G, H (not E-H; K, L (not I-L); O-Q (not M-Q), and T, U (not R-U).

Response:We appreciate for your comments and corrected as suggested.

Figure 8, 271-285 lines. The description could be shortened by avoiding the repetitive phrase “the in situ hybridization analysis” (repeated 3 times).

Response:Thanks to your comments, we have deleted the repetitive " the in situ hybridization analysis of" in lines 271-285 to keep the manuscript concise.

Bar=100nm? Is it true?

Response:Thank you very much for your comments, and we are sorry that we mistakenly marked the correct Bar "µm" in the legend as the wrong "nm", which has been revised in line 288.

Discussion

The discussion is difficult to read in many parts.

In the first part, 3.1, the Discussion greatly overlaps with the Results. Perhaps, some schematic representation would be helpful.

Response:We appreciate for your comments. We reorganized and revised the discussion section, and added a schematic diagram to help with understanding.

In the second part, 3.2, 363-375 lines, the text is not quite relevant to this study and needs to be modified.

Response:Thank you very much for your comments. We have modified the second part of the discussion as suggested.

Conclusions

It needs to be thoroughly checked through for the correct use of the terms “dorsal and ventral domains, parts…petals”. An example of mixing these terms is the sentence in lines 418-420, “In tubular types CmCYC2c was sustained symmetric expressed in the dorsal and ventral domain of ray petals, whereas in the flat types, it was almost no expression in the dorsal petals and expression in the ventral petals.”

Response:We are grateful to this comments. We checked through the manuscript and corrected the confused terms. We used the “dorsal and ventral domains of ray petals”, “dorsal domain” and “ventral domain” to replace other similar terms.

Reviewer 2 Report

Comments and Suggestions for Authors

The manuscript holds significance in enhancing our comprehension of the morphology and expression patterns of CmCYC2c in chrysanthemums (Chrysanthemum morifolium) across various floral forms. During my review, I identified a few minor issues that require correction:

1. Lines 35-36: Authors should add the references.

2. Lines 44-46: Authors should add the references.

3. Lines 51-53: Authors should add the references.

4. Figure 2: Authors should correct the petal stage of Figure 2D. I think it is not correct.

5. Author should describe the molecular mechanisms responsibility of the differential expression of CmCYC2c between ray florets and disc florets.

6. How does these differential expressions contribute to the distinct petal shapes and capitulum forms observed in chrysanthemum cultivars with varying petal types?

7. How do the findings regarding the spatiotemporal expression patterns of CmCYC2c in different types of ray flowers, particularly the variations in the dorsal and ventral domains of ray petals, inform your understanding of the genetic control of petal morphogenesis and its role in shaping the diversity of flower types in chrysanthemum? Author should discuss based on my suggestion.

  Comments on the Quality of English Language

This manuscript holds significant value for researchers and enthusiasts studying chrysanthemums. It is imperative to address necessary corrections and delve into a comprehensive discussion of CmCYC2 expression, particularly its impact on the flowering stage, with a specific focus on stage 4.

Author Response

Comments and Suggestions for Authors

The manuscript holds significance in enhancing our comprehension of the morphology and expression patterns of CmCYC2c in chrysanthemums (Chrysanthemum morifolium) across various floral forms. During my review, I identified a few minor issues that require correction:

  1. Lines 35-36: Authors should add the references.

Response:Thank you very much for your suggestions, in lines 36-37 two references have been added.

‘Teixeira da Silva JA: Chrysanthemum: advances in tissue culture, cryopreservation, postharvest technology, genetics and transgenic biotechnology. Biotechnol Adv 2003, 21(8):715-766.’   ‘Anderson NO: In: Flower breeding and genetics.: Dordrecht: Springer Netherlands; 2006.p.389-437.’

  1. Lines 44-46: Authors should add the references.

Response:Thanks for your comments and the reference has been added.

“Fan J, Huang J, Pu Y, Niu Y, Zhang M, Dai S, Huang H: Transcriptomic analysis reveals the formation mechanism of anemone-type flower in chrysanthemum. BMC Genomics 2022, 23(1):846.”

Pu Y, Huang H, Wen X, Lu C, Zhang B, Gu X, Qi S, Fan G, Wang W, Dai S: Comprehensive transcriptomic analysis provides new insights into the mechanism of ray floret morphogenesis in chrysanthemum. BMC Genomics 2020, 21(1):728.

  1. Lines 51-53: Authors should add the references.

Response:Thanks for your comments and the reference has been added.

 “Zhang Y, Dai S, Hong Y, Song X: Application of genomic SSR locus polymorphisms on the identification and classification of chrysanthemum cultivars in China. PLoS One 2014, 9(8):e104856.”

  1. Figure 2: Authors should correct the petal stage of Figure 2D. I think it is not correct.

Response:Thank you very much for your comments. In this picture, the right panel is ray florets from outside to inside whorls, not showing the florets at different stages. To avoid misunderstanding, we have added it in the figure legends. The size of the ray florets of ‘Baipingpong’ is very uniform from the inside to the outside, thus forming pompon type.

  1. Author should describe the molecular mechanisms responsibility of the differential expression of CmCYC2c between ray florets and disc florets.

Response:Thank you very much for your comments. We added the discussion of the molecular mechanisms responsibility of the differential expression of CmCYC2c between ray florets and disc florets in line 384-402.

  1. How does these differential expressions contribute to the distinct petal shapes and capitulum forms observed in chrysanthemum cultivars with varying petal types?

Response:We are grateful to this comments. We added the relevant discussion in the line 403-408

  1. How do the findings regarding the spatiotemporal expression patterns of CmCYC2c in different types of ray flowers, particularly the variations in the dorsal and ventral domains of ray petals, inform your understanding of the genetic control of petal morphogenesis and its role in shaping the diversity of flower types in chrysanthemum? Author should discuss based on my suggestion.

Response:We appreciate for your suggestions. We added the relevant discussion in the line 403-415.

Comments on the Quality of English Language

This manuscript holds significant value for researchers and enthusiasts studying chrysanthemums. It is imperative to address necessary corrections and delve into a comprehensive discussion of CmCYC2 expression, particularly its impact on the flowering stage, with a specific focus on stage 4.

Response:We appreciate for your suggestions. We added the relevant discussion in the line 403-415.